# Simulation-Based Design of a Low-Cost Broadband Wide-Beamwidth Crossed-Dipole Antenna for Multi-Global Navigational Satellite System Positioning

**DOI:** 10.3390/s25154665

**Published:** 2025-07-28

**Authors:** Songyuan Xu, Jiwon Heo, Won Seok Choi, Seong-Gon Choi, Bierng-Chearl Ahn

**Affiliations:** 1School of Intelligent Manufacturing, Sichuan University of Arts and Science, Dazhou 635000, China; 20250112@sasu.edu.cn; 2School of Electric and Computer Engineering, Chungbuk National University, Cheongju 28644, Republic of Korea; heo1234@chunbuk.ac.kr (J.H.); wschoi@chungbuk.ac.kr (W.S.C.)

**Keywords:** crossed dipole, multi-GNSS, wide beamwidth, dual polarization

## Abstract

This paper presents the design of a wideband circularly polarized crossed-dipole antenna for multi-GNSS applications, covering the frequency range of 1.16–1.61 GHz. The proposed antenna employs orthogonally placed dipole elements fed by a three-branch quadrature hybrid coupler for broadband and wide gain/axial ratio beamwidth. The design is carried out using CST Studio Suite for a single dipole antenna followed by a crossed-dipole antenna, a feed network, and the entire antenna structure. The designed multi-GNSS antenna shows, at 1.16–1.61 GHz, a reflection coefficient of less than −17 dB, a zenith gain of 3.9–5.8 dBic, a horizontal gain of −3.3 to −0.2 dBic, a zenith axial ratio of 0.6–1.0 dB, and horizontal axial ratio of 0.4–5.9 dB. The proposed antenna has a dimension of 0.48 × 0.48 × 0.25 *λ* at the center frequency of 1.39 GHz. The proposed antenna can also operate as an LHCP antenna for *L*-band satellite phone communication at 1.525–1.661 GHz.

## 1. Introduction

The Global Navigation Satellite System (GNSS) is continuously evolving with improvements in hardware and algorithms. Starting from the original GPS technology, it is now a standard practice to utilize all of the available GNSS signals from GLONASS, Galileo, and BeiDou satellites [1,2]. The use of multi-GNSS signals on a range of different frequencies offers many advantages: optimization of satellite geometry for reduced PDOP (Position Dilution of Precision) [3,4]; improved positioning reliability when the sky is blocked by tall buildings in urban canyons, by trees in dense forests, by steep mountains, and by terrain in deep valleys; reduced positioning error through ionospheric and tropospheric delay estimation. The multi-GNSS approach enables advanced techniques such as PPP (Precise Point Positioning), RTK (Real-Time Kinematic), and PPP-RTK and improved robustness against jamming and spoofing [5,6,7]. The multi-GNSS-based positioning and time dissemination give rise to new applications such as survey [8], natural hazard detection [9,10], weather prediction and meteorological studies [11,12], infrastructure monitoring [13], geology and Earth science [14,15], vehicle self-driving [16], and precision agriculture [17].

These technological developments also pose significant challenges to GNSS hardware design, particularly for antennas. High-performance GNSS antennas must now meet rigorous requirements, including support for multiple constellations and frequency bands [18,19,20]. For highly maneuvering or dynamic platforms such as military ground vehicles and aircraft, marine vessels, and land vehicles moving on rough terrain, it is advantageous to use GNSS antennas with wide gain and axial ratio beamwidth to offset the effect of platform movements [21].

Wide-beamwidth antennas offer such advantages as reduced PDOP, early signal acquisition from rising satellites, and reduced loss of signal reception and cycle slips [22,23]. In designing GNSS antennas with a wide beamwidth on board highly maneuvering platforms, it is important to maintain required the gain and axial ratio (AR) from the zenith down to the horizon for reliable reception of GNSS signals.

The paper presents a simulation-based design of a broadband wide-beamwidth crossed-dipole antenna for worldwide GNSS signal reception. There have been some research works on multi-GNSS antennas with broad gain and axial ratio beamwidths. Li and co-workers presented a GNSS antenna consisting of two bow-tie crossed dipoles on their PCB (printed circuit board) supported by four metal posts, a partial-height metal cavity, and a feed network consisting of a Wilkinson power divider and a 90-degree phase-delay line. Their dipole surface is parallel to the ground plane [21]. They achieved an axial ratio beamwidth of 130°–150° and a half-power beamwidth of 115°–125° at 1.227–1.575 GHz. Choi and co-workers designed a wide beamwidth antenna using bent bow-tie metal-plate dipoles with a coaxial balun fed by a two-stage Wilkinson power divider and a 90-degree phase-delay line [22]. They employed a large circular ground plane with a diameter of 1.05 *λ* and achieved a wide beamwidth by placing the crossed dipoles at 0.55 *λ* above and parallel to the ground plane. Li and co-workers designed an RHCP antenna operating at 0.28–0.40 GHz with a wide axial ratio beamwidth [23]. They employed curved printed crossed dipoles fed by a Wilkinson power divider combined with a phase-delay line.

Zhang and co-workers designed a single-band GNSS antenna with a wide axial ratio beamwidth employing semicircular wire crossed dipoles fed in phase quadrature using a quarter-wavelength coaxial cable [24]. Their ground plane is large (0.85 *λ*) so that the dipole is placed high (0.42 *λ*) above the ground plane. Yang and a co-worker proposed an ultra-low profile design (0.1 *λ*) with a 3 dB axial ratio beamwidth > 120°, yet it exhibits limited low-elevation gain and is primarily suited for static scenarios [25]. They employed crossed bow-tie dipoles fed by a coaxial cable with an integrated 90-degree phasing ring. Four parasitic triangular patches are placed between dipole arms and shorted to a square ground plane. Sun and co-workers presented low-elevation gain (0.11 dBic at *θ* = 85°) using curved crossed dipoles in a corrugated back cavity and achieved a 3 dB axial ratio beamwidth in excess of 200°, but the design is complex and narrow band so that it can be used only for GNSSs at the 1.6 GHz band [26]. A circularly polarized antenna radiating isotropically at all angles has a gain of 0 dBic, where the subscript ‘ic’ means ‘relative to isotropic circularly polarized radiator’.

This paper proposes a novel wideband and broad-beamwidth circularly polarized crossed-dipole antenna tailored for collaborative multi-GNSS reception on highly maneuvering platforms. The core design objective of the proposed antenna is to obtain acceptable gain and axial ratio performance in the entire hemispherical angles at 1.16–1.61 GHz. The design goals are achieved by employing vertically arranged printed crossed dipoles with parasitic monopoles, a circular ground plane of optimum diameter, and a three-branch-line quadrature hybrid coupler realized on the back side of the ground plane. The designed antenna exhibits a gain > −3.3 dBic and an axial ratio < 6 dB within ±90° elevation, or in the upper hemisphere, with a peak gain of 5.8 dBic. The proposed design provides broad beamwidth for multi-GNSS signal reception on attitude-changing platforms.

In the following, we present a detailed description of the design of the proposed antenna conducted using CST Studio Suite, which accurately predicts the antenna’s performance, eliminating the immediate need for physical prototypes. In Section 2, the design specifications and the antenna structure are described, followed by the design of a single dipole antenna in Section 3. Section 4 and Section 5 deal with the design of the crossed-dipole antenna and the antenna feed network, respectively. The final design and performance of the proposed antenna are presented in Section 6, followed by the conclusion of the paper in Section 7. We present, in great detail, the design steps and intermediate results along with the dimensions of the final design so that the proposed antenna can easily be implemented even by laymen in the general technical community.

## 2. Design Specifications

In this section, we present the design specifications and the structure of the proposed antenna. Table 1 summarizes the design goal of our multi-GNSS antenna. The first six parameters, from frequency range to axial ratio, are essential performance criteria required of a GNSS antenna. Antenna size is typical of a single antenna that can meet design specifications in Table 1. GNSS RF signals are transmitted in the RHCP (right-hand circular polarization). The frequency range of 1.16–1.61 GHz is set to cover the worldwide GNSS currently in operation (1.164–1.300 GHz and 1.559–1.610 GHz).

For impedance-matched performance, a reflection coefficient of less than −10 dB is specified. For wide beamwidth operation, the minimum antenna gain at zenith (*θ* = 0°) needs to be relaxed to 3 dBic. The minimum antenna gain at the horizon (*θ* = 90°) is set to −4 dBic, reflecting the physical realizability in a given dimension. An omnidirectional gain of 3 dBic over the hemispherical space and no radiation (minus infinite gain) below the horizon can be obtained only from an ideal imaginary antenna, which occupies an infinitely large space. The axial ratio of less than 6 dB is specified over the upper hemispherical angles (−90° ≤ *θ* ≤ 90° and −90° ≤ *φ* ≤ + 90°), not just on the principal planes. This value is also selected, reflecting the realizability. The normal axial ratio of 3 dB is difficult to achieve at all angles in the upper hemisphere so that it is relaxed to 6 dB. Assuming a 1 dB axial ratio for the GNSS transmitting antenna onboard a satellite, the former corresponds to 0.23 dB in the maximum polarization mismatch loss, while the latter corresponds to 0.64 dB [27].

Antenna types capable of meeting design specifications in Table 1 include crossed dipoles [21,22,23,24,25,26], four-point fed patches [28], four horizontal arc monopoles [29], and quadrifilar helices [30]. In this work, we choose wideband crossed-dipoles with parasitic monopoles fed by a broadband quadrature hybrid coupler realized on the dipole ground plane of optimum size. This structure requires only three printed circuit boards (PCBs) that can be easily assembled by inserting PCBs in slots and using soldering PCBs at suitable locations. The proposed antenna with a wide beamwidth is amenable to low-cost production and, for example, can be attractive for small fishing boats that can constantly undergo rolling and pitching.

## 3. Antenna Structure

Figure 1 shows an integrated 3D layout of the proposed antenna. It consists of two orthogonal printed dipole antennas (Dipole *A* and Dipole *B*) fed by a broadband quadrature hybrid coupler (QHC) printed on the backside of the ground plane circuit board. To obtain circularly polarized radiation, it is necessary to excite two orthogonal current vectors in phase quadrature. The electric field at the far zone can be expressed in the spherical coordinates as follows [31].(1)$$E=E_{\theta }\hat{\theta }+E_{\varphi }\hat{\varphi }=E_{R}a_{R}+E_{L}a_{L}$$
where *E_θ_* and *E_φ_* are the phasor of the theta and phi components of the electric field, and **a***_R_* and **a***_L_* are base vectors for the RHCP and LHCP, respectively, that are given by the following:(2)$$a_{R}=\frac{1}{\sqrt{2}}(\hat{\theta }-j\hat{\varphi })\;,\;a_{L}=\frac{1}{\sqrt{2}}(\hat{\theta }+j\hat{\varphi })$$

*E_R_* and *E_L_* are the RHCP and LHCP components of the electric field given by the following:(3)$$E_{R}=E\cdot a_{R}^{*}=\frac{1}{\sqrt{2}}(E_{\theta }+jE_{\varphi })\;,\;E_{L}=E\cdot a_{L}^{*}=\frac{1}{\sqrt{2}}(E_{\theta }-jE_{\varphi })$$

The maximum electric field or the semi-major axis of the polarization ellipse is given by the following:(4)$$\left|E{|}_{\max}=\frac{1}{\sqrt{2}}(|E_{R}|+|E_{L}|\right)$$

While the minimum electric field or the semi-minor axis is given by the following:(5)$$|E{|}_{\min}=\left\{\begin{array}{l} \frac{1}{\sqrt{2}}(|E_{R}|-|E_{L}|)\;\mathrm{if}\;|E_{R}|\;>\;|E_{L}|\;(\mathrm{RHCP})\\ \frac{1}{\sqrt{2}}(|E_{L}|-|E_{R}|)\;\mathrm{if}\;|E_{L}|\;>\;|E_{R}|\;(\mathrm{LHCP}) \end{array}\right.$$

The axial ratio (AR) is now given by the following:(6)$$\mathrm{AR}=20\log_{10}\frac{|E{|}_{\max}}{|E{|}_{\min}}$$

For an ideal axial ratio of 0 dB, one needs two orthogonal electric field vectors of the same magnitude and of ±90° phase difference, where the plus or minus sign determines the sense of polarization, i.e., RHCP or LHCP. Two symmetric crossed dipoles fed in 90-degree phase difference in the absence of a conducting ground plane offer 0 dB axial ratio in the direction normal to the plane of the dipoles and infinite axial ratio in the plane of the dipoles.

One method of obtaining a wide axial ratio and gain beamwidths is to place the crossed dipoles at a suitable distance from a ground plane of optimum diameter and to add parasitic monopoles excited by the dipoles as shown in Figure 1. The monopoles generate an electric field vector normal to the ground plane, which widens the axial ratio and gain beamwidths. The size of the ground plane is important since a large ground plane will short-circuit the electric field vector tangential to the ground plane, leaving out only the normal component of the electric field. This in turn makes the axial ratio approach infinity.

For wide beamwidth over a large frequency range, the dimensions of the dipoles, monopoles, and ground plane are optimized along with the dipole feed network using an electromagnetic simulation tool. For a good axial ratio, the balancing of the dipole current and the monopole current is important as well as the size of the ground plane. For broadband performance, the width of the dipole strip and the shape of the monopole are key parameters, as well as the wideband feed network. Table 2 shows the axial ratio versus the amplitude ratio and phase difference in two vector components, *E_θ_* and *E_φ_*. The axial ratio is calculated and tabulated using Equation (6) for the phase difference (Arg (*E_θ_*)–Arg (*E_φ_*)) from 50 to 90 degrees and the magnitude ratio (|*E_θ_*|/|*E_φ_*|) of 0, 1, 2, and 3 dB. For an amplitude ratio of less than 2 dB, a deviation of 15 degrees from the 90° phase difference can be tolerated for an axial ratio of less than 3 dB.

The proposed antenna is designed using an FR-4 substrate with a dielectric thickness of 1.0 mm, a relative permittivity of 4.3, a loss tangent of 0.025, and a conductor thickness of 0.034 mm. Two crossed dipoles are excited in phase quadrature for circular polarization. The output terminals of the QHC are connected to two coaxial cables, one for RHCP and the other for LHCP (left-hand circular polarization) operation. The dipole arms are printed on two orthogonally oriented vertical circuit boards, which host a microstrip feed line with an integrated balun as well. Two parasitic monopoles of trapezoidal shape are printed under the dipole arm to improve the symmetry of the dipole’s *E*- and *H*-plane gain patterns, thus improving the axial ratio, and to increase the antenna gain at low elevation angles in the dipole’s *E*-plane. The short-circuited part of the slotline and the monopole are electrically connected at the base to the ground plane of a circular printed circuit board (PCB).

In a real implementation, to accurately position the dipole PCBs, small rectangular via holes can be constructed on the ground plane PCB such that the via hole width is slightly larger than the thickness of the dipole PCB. Metal-clad legs with dimensions corresponding to the rectangular via hole are added to the bottom of the dipole PCB. Mechanical fastening of the ground plane PCB and the dipole PCB can be performed by inserting the metal-clad legs in the rectangular via holes and soldering them at suitable points. These structures are not shown in Figure 1 for clarity.

In feeding a dipole, it is important to make currents on two arms of the dipole be balanced (the same magnitude and phase). This is performed by a microstrip-to-slotline transition with an integrated balun. The quadrature-phase feeding is enabled by using a broadband QHC, whose output arms are connected to the input of the dipole feed line. The QHC enables the feeding of the orthogonal dipole arms with equal magnitude and a 90-degree phase difference. A three-branch-line type is employed to obtain a bandwidth large enough for the proposed antenna.

A metal coating on the top side of the circular PCB provides a ground plane (GP) for the dipole/monopole operation as well as that for the QHC. A circular-shaped GP is chosen for the angular symmetry of the gain and axial ratio pattern. The diameter of the GP is critical in the realization of the wide axial ratio beamwidth since a large-sized GP will short-circuit the tangential component of the dipole’s electric field while doubling the vertical component, resulting in the deterioration of the axial ratio.

A self-assembled PCB-based structure shown in Figure 1 of the dipole, monopole, feed line, and QHC results in a compact low-cost antenna. Compared with some of the existing designs, the following are among the novelties of the proposed antenna:(1)The antenna provides decent gain and axial ratio (AR) performance over the entire angle in the upper hemisphere at frequencies covering all of the worldwide GNSS services currently in operation.(2)No additional structure for the assembly of the dipole PCB and the feed network PCB is required.(3)The design employs a simple structure amenable for low-cost fabrication.(4)It offers the ability of dual-purpose use for worldwide GNSS (RHCP) and *L*-band satellite phone (LHCP; 1.525–1.661 GHz; Iridium, Inmarsat, and Thuraya).

Existing works do not present gain and AR characteristics at all angles in the upper hemisphere. Therefore, uncertainties remain regarding the performance of existing designs in the entire upper hemisphere.

The overall structure has been explained above using Figure 1. Detailed structural configurations of the proposed antenna are illustrated in Figure 2 and Figure 3. Figure 2 shows the two dipoles and associated monopoles along with the feed lines. The dipole is fed by a microstrip line to slotline transition and an integrated balun. The microstrip line mode is converted to the slotline mode by coupling between the slot with a strip. The width of the microstrip line in the coupling region is halved so the feed point of Dipole *A* and Dipole *B* is as close to each other as possible. This ensures nearly identical performance of the two dipoles.

The microstrip feed line is terminated in an open circuit at a quarter-wave point, while the slotline is short-circuited at a quarter-wave distance from the microstrip–slotline coupling point. This structure is geometrically as symmetric as possible, resulting in the balanced excitation of the current on the two arms of the dipole. The combination of a quarter-wave open-circuited microstrip line and a quarter-wave short-circuited slotline yields a broadband transition between a microstrip line and a slotline. This idea of feeding a printed dipole was first proposed by Edward and Rees in 1987 [32] and has widely been employed in feeding the broadband printed dipole antenna [33,34,35,36,37].

To assemble two dipole PCBs in a cross configuration, a cutaway slot is formed on the Dipole *A* PCB and on the Dipole *B* PCB. The width of the cutaway slot is slightly larger than the thickness of the PCB. The cutaway slot ranges from just above the microstrip line to the top edge in the Dipole *A* PCB, while it ranges from the bottom edge to just below the microstrip line. The slotline of Dipole *A* is short-circuited at a quarter-wavelength distance from the slot–strip coupling point, while the short-circuiting of the slotline of Dipole *B* is realized using a via on the PCB of Dipole *A*. Wideband impedance matching is possible by a proper choice of the position of the slot–strip coupling point and by the combined frequency characteristics of the slotline short circuit and the microstrip line open circuit. The resulting balun structure provides the balanced feeding of the dipole over a broad bandwidth.

Structures for the interconnection of the antenna components are highlighted in Figure 3 through magnified drawings. The box 1 in Figure 3a denotes the short-circuiting of the slotline of Dipole *B* using a via on the PCB of Dipole *A*. It is presented in a magnified drawing of Figure 3b, where via pads in the Dipole *A* PCB are soldered to the slotline conductor in the Dipole *B* PCB. Via and via pads on each side of the dipole *A* PCB can be identified. The short-circuiting of the slotline balun for Dipole *B* can be completed by soldering the via pad on the Dipole *A* PCB to the conductor trace on both sides of the slotline of Dipole *B*. In simulation, the soldering is emulated by joining the two metal surfaces.

The slot width of the slotline above the via In the Dipole *B* PCB is about three times wider than the one below the via. This is performed to prevent the slotline in the Dipole *B* PCB from touching the slotline in the Dipole *A* PCB. The slot width of the slotline below the via is about the same as the thickness of the Dipole *A* PCB, as shown in Figure 2b.

The box 2 in Figure 3a is the structure for connecting the output of the QHC to the input of the dipole feed line. Two vias are utilized to connect the output of the quadrature hybrid coupler to the microstrip line feeding the dipole. A small rectangular hole is formed in the ground plane PCB for electrical insulation between the via pad and the ground plane conductor. The via pad can be electrically connected to the microstrip line by soldering.

The box 3 in Figure 3a shows a method for connecting an input of the QHC to a coaxial cable. Its magnified view is presented in Figure 3d. The RHCP port of the QHC input is connected to one coaxial cable, while the other input port (for LHCP operation) is connected to another coaxial cable for the *L*-band satellite phone application. The coaxial center conductor is routed to the input end of the quadrature hybrid coupler through a via, while the coaxial outer conductor is electrically connected to the ground plane (GP) conductor on the top surface of the GP PCB. Again, soldering can be used for the electrical connection of conductor surfaces.

Figure 4a shows the microstrip QHC realized on the backside of the GP PCB, while Figure 4b shows the vias and the via pads on the top side of the GP PCB. To achieve an impedance bandwidth large enough for worldwide GNSS signal reception, a three-branch-line QHC is employed. To reduce the size, the main lines are meandered. The output lines of the QHC are routed in a circular shape to the position of the dipole feed line.

## 4. Design of a Single Dipole Antenna

In this section, we present the design of a single dipole antenna with parasitic monopoles. This step is most important since the bandwidth and beamwidth characteristics are largely determined by the dipole antenna and parasitic monopoles. Figure 5 shows the structure used in the design of a single dipole. Dimensions of Dipole *A* are determined by iterative parametric analysis. Dipole *B*’s dimensions are the same as those of Dipole *A* in Figure 5a except for the slot–microstrip coupling region. Dipole *B* in Figure 5b is presented for the purpose of comparison with Dipole *A*. Using only a single element in the crossed dipole design is valid since the presence of the other orthogonally placed dipole has only a small effect due to a large isolation between the two dipoles. In the simulation, an infinitesimal or delta-gap current source (‘Discrete Port’ in CST Studio) is used between the input end of the microstrip line and the ground plane.

Figure 6 shows the dimensional parameters in the dipole design. Figure 6a shows a plan view of two crossed dipoles installed on a circular ground plane along with *x* and *y* axes of the rectangular coordinate system for far-field pattern angle reference. Figure 6b–d shows the dimensional parameters of the dipole and the feed line. Dimensional parameters of Dipole *A* without Dipole *B* are iteratively adjusted using the structure of Figure 5a for broadband impedance matching and rotationally symmetric gain patterns over large elevation angles. When Dipole *B* is added to form crossed dipoles, its dimensions are set identical to those of Dipole *A* except for *C*_1*B*_, the position of the slot–strip coupling.

The dipole length *L* determines the center frequency of operation, while the dipole width *A*_1_ determines the bandwidth. The value of *A*_1_ should be large enough for broadband operation. The gain pattern in the elevation plane is sensitive to the dipole height from the ground plane, whose value is to be iteratively adjusted while monitoring the elevation gain pattern over the frequency range of operation. In the initial stage, the diameter of the ground plane is made the same as the length of Dipole *A* and is later adjusted together with crossed Dipole *A* and Dipole *B* for the wide axial ratio beamwidth.

The use of the monopole is critical to achieving a rotationally symmetric gain pattern since with a dipole alone the gain pattern will be highly asymmetric in *E*- and *H*-planes, resulting in a large axial ratio. The dipole height (*A*_2_ + *A*_3_) determines the height *A*_3_ of the monopole. The current in the parasitic monopole is induced by capacitive coupling via the gap *A*_2_. For optimum coupling, the gap *A*_2_ and the width *A*_4_ are iteratively adjusted. The base width *B*_3_ of the monopole needs to be large enough for broadband operation. The position of the monopole’s vertical edge is relatively insensitive to the impedance matching, gain pattern, and axial ratio as far as it is near the end of the dipole arm. Therefore, the monopole is placed such that the monopole’s vertical edge is aligned with the dipole tip.

The dipole is fed by using a 50-ohm microstrip line with an integrated balun. The slot width *G* determines the characteristic impedance of the slotline and is not sensitive to the impedance matching as far as it is not too large, so it is set to 3.0 mm, three times the substrate thickness. With this value, the characteristic impedance of the uniform slotline, i.e., without the other dipole’s substrate, is 132 Ω. With the other dipole’s PCB inserted in the slot, the characteristic impedance will be decreased due to dielectric loading.

The heights *H_A_* (for Dipole *A*) and *H_B_* (for Dipole *B*) determine the short-circuited line length of the slotline together with the slot–strip coupling point *C*_1*A*_ (for Dipole *A*) and *C*_1*B*_ (for Dipole *B*). The position *C*_1*A*_ of the microstrip–slotline coupling point relative to the arm of Dipole *A* is crucial in the impedance matching, and it is adjusted by parameter sweep. When crossed dipoles are formed in a later stage, the corresponding value *C*_1*B*_ in Dipole *B* is set as close to *C*_1*A*_ as possible to excite the two dipoles with almost the same phase and to obtain the same impedance matching. The gap between the half-width microstrip line crossing the slotline is set to 0.5 mm. The coupling position *C*_1*A*_, position *H_A_* of the slotline short circuit, and length *C*_5_ of the microstrip open circuit are all interrelated and thus changed simultaneously.

The dipole design is the most laborious step in the development of the proposed antenna, requiring extensive simulations as well as intuition and experience. The results of the design of Dipole *A* are presented in Table 3 for dimensions, in Figure 7 for reflection coefficient, and in Figure 8 for gain patterns. Dimensions in Table 3 are values slightly readjusted with the addition of Dipole *B* and the quadrature hybrid coupler as described in the later sections.

Figure 7 shows the reflection coefficient of the designed Dipole *A* and Dipole *B* when they are alone on a ground plane as shown in Figure 5. The reflection coefficient of Dipole *A* is less than −10 dB from 1.10 GHz to 1.74 GHz with resonances at 1.20 GHz and 1.56 GHz. The resonance at 1.20 GHz is due to the dipole, while the one at 1.56 GHz is due to the monopole. The resonance frequency is determined by the combined effects of the dipole, monopole, dipole-monopole coupling, ground plane diameter, and dipole feed line. From Table 3, the lengths of the dipole and the monopole in terms of the resonant wavelength at 1.20 GHz and 1.56 GHz are found to be 0.424 *λ* and 0.179 *λ*, respectively. The reflection coefficient of Dipole *B* shows a frequency response similar to Dipole *A*.

Figure 8 shows the co-polarized gain patterns of the design Dipole *A* at 1.16, 1.30, and 1.61 GHz for *φ* = 0°, 45°, 90°, and 135° with Dipole *A* in the *φ* = 135° plane as shown in Figure 6a. First, it is noted that the *E*-plane (*φ* = 135°) pattern is still narrower than the *H*-plane (*φ* = 45°) pattern. The two diagonal plane (*φ* = 0°, 90°) patterns are the same. The beamwidth is increased as the frequency increases. This is due to the fact that the dipole resonates at 1.20 GHz while the monopole resonates at 1.56 GHz. At lower frequencies, the dipole current is stronger, resulting in decreased gain in the horizontal angle (*θ* = 90°) and increased gain at zenith (*θ* = 0°). At higher frequencies, the monopole is more strongly excited, yielding reduced zenith gain and increased horizontal gain. With the use of parasitic monopoles, the angular symmetry of the gain pattern in the *φ* direction is much improved from that of a dipole alone. The front-to-back ratio ranges from 10 dB to 15 dB.

## 5. Design of a Crossed-Dipole Antenna

With the design of a single dipole antenna completed, Dipole *B* is constructed, whose dimensions are the same as those of Dipole *A* except for the microstrip–slotline coupling point position *C*_1*B*_. Dipole *A* and Dipole *B* are arranged in orthogonal directions as shown in Figure 9 to form a crossed-dipole antenna. Rectangular coordinate axes are shown in Figure 9 for far-field angle identification in spherical coordinates. Dipoles *A* and *B* are placed in *φ* = 45° and *φ* = 135° planes, respectively. Dimensions of Dipole *A* and Dipole *B* are slightly readjusted for good impedance matching and high isolation between the two dipoles. The ground plane diameter is finely tuned for wide gain and axial ratio beamwidths. Dipole *A* and Dipole *B* are excited with equal magnitude and with a 90° phase difference using delta-gap sources. This is equivalent to feeding the dipoles with a perfect quadrature hybrid coupler.

Dimensions of the designed crossed-dipole antenna are given in Table 3. Figure 10 shows the reflection coefficient of Dipole *A* (S1,1) and Dipole *B* (S2,2) and the transmission coefficient (S2,1) from Dipole *A* to Dipole *B*. The transmission coefficient S1,2 from Dipole *B* to Dipole *A* is not drawn in Figure 10 since it is the same as S2,1. The reflection coefficient of Dipole *B* is similar to that of Dipole *A* and less than −10 dB above 1.10 GHz. The transmission coefficient, or negative of isolation, is less than −24 dB above 1.10 GHz.

Figure 11 shows 2D RHCP gain patterns of the crossed-dipole antenna at 1.16, 1.30, and 1.61 GHz. In the spherical coordinate system, the range of angles *θ* and *φ* can be defined in various ways. In Figure 11, the elevation angle *θ* (Theta) ranges from −180° to +180°, while the azimuth angle *φ* (Phi) ranges from −90° to +90°, covering the entire angle range in the spherical coordinate. Angular symmetry of gain patterns in the *φ* direction is excellent. From 1D gain patterns (not shown), we find that the maximum gain *G*_max_ and the minimum gain *G*_min_ in the horizontal plane are as follows:

At 1.16 GHz, *G*_max_ = −2.65 dBic at *φ/θ* = 62°/90° and *G*_min_ = −3.06 dBic at *φ/θ* = −48°/−90°;At 1.30 GHz, *G*_max_ = −1.42 dBic at *φ/θ* = 36°/−90° and *G*_min_ = −2.61 dBic at *φ/θ* = −73°/90°;At 1.61 GHz, *G*_max_ = −0.55 dBic at *φ/θ* = 34°/−90° and *G*_min_ = −1.22 dBic at *φ/θ* = 78°/90°.

The horizontal gain is increased with frequency, as can be seen from the above numerical values and Figure 11.

Figure 12 shows the 2D axial ratio (AR) patterns of the crossed-dipole antenna at 1.16, 1.30, and 1.61 GHz. Axial ratio is less than 6 dB in the upper hemisphere (−90° ≤ *θ* ≤ +90° and −90° ≤ *φ* ≤ +90°). In the 1D gain patterns of a single dipole antenna shown in Figure 8, one can see that the gain symmetry versus the *φ* angle is better at 1.16 GHz than at 1.30 GHz, suggesting a better AR performance at 1.16 GHz. This can be seen by comparing Figure 12a with Figure 12c. The AR is largest at the horizon (*θ* = 90°) in the upper hemisphere (−90° ≤ *θ* ≤ +90°). From the analysis of 1D AR patterns (not shown), we find the AR at the horizon as follows:

At 1.16 GHz, AR_max_ = 4.74 dB at *φ/θ* = −50°/−90° and AR_min_ = 2.14 dB at *φ/θ* = 12°/−90°;At 1.30 GHz, AR_max_ = 5.92 dB at *φ/θ* = −43°/−90° and AR_min_ = 0.95 dB at *φ/θ* = 15°/−90°;At 1.61 GHz, AR_max_ = 4.88 dB at *φ/θ* = −64°/−90° and AR_min_ = 0.07 dB at *φ/θ* = 17°/90°.

## 6. Design of a Feed Network

To cover the worldwide GNSS frequencies, a feed network for the crossed-dipole antenna should provide two balanced outputs with a 90° phase difference over a broad bandwidth. In this work, we employ a three-branch line microstrip quadrature hybrid coupler (QHC), which is printed on the backside of the circular ground plane. The use of the QHC enables the antenna to be used for the LHCP *L*-band satellite phone (1.525–1.661 GHz) as well.

The two-branch line coupler has a bandwidth of 15–20%, which is not sufficient to cover 1.16–1.61 GHz (32.5% bandwidth) [38], so we employ a three-branch line coupler. Starting with an unreduced-size QHC shown in Figure 13a, a reduced-size QHC shown in Figure 13b is designed. The QHC of Figure 13a is designed at the center frequency of 1.39 GHz for Butterworth response with the following theoretical parameter values [39]: *Z*_1_ = 120.5 Ω, *Z*_2_ = 37.2 Ω, *Z*_3_ = 36.3 Ω, *L*_1_ = 0.249 *λ*, *L*_2_ = 0.264 *λ*, and *L*_3_ = 0.270 *λ*, where *λ* is the wavelength of the corresponding microstrip lines. This design gives 2.4 dB and 3.3 dB differences in transmission coefficients at 1.16 GHz and 1.61 GHz, respectively. To improve the magnitude balance, line widths and lengths are optimized, resulting in the following values: *Z*_1_ = 99.7 Ω, *Z*_2_ = 31.2 Ω, *Z*_3_ = 62.0 Ω, *L*_1_ = 0.253 *λ*, *L*_2_ = 0.265 *λ*, and *L*_3_ = 0.261 *λ*. The improved design shows 0.33–0.38 dB in amplitude balance and 88.9°–90.5° in phase difference at 1.16–1.61 GHz.

The reduced QHC shown in Figure 13b is obtained from the optimized design of the QHC shown in Figure 13a by meandering the main lines so that the coupler’s size can be fitted within the boundary of the crossed-dipole antenna’s ground plane. The line widths and lengths are readjusted after meandering. In Figure 13b, Port 1 and Port 4 are the RHCP and LHCP inputs, respectively, while Port 2 is the coupled output port (connected to the Dipole *B* feed line) with a 0° phase reference, and Port 3 is the coupled output (connected to the Dipole *A* feed line) with a −90° phase with respect to the 0° phase reference. Note that the QHC shown in Figure 1 is the one seen from the top side of the ground plane, while the QHC of Figure 13b is the one seen from the back side of the ground plane. We can call Port 2 and Port 3 the in-phase port and the quadrature-phase port, respectively.

Table 4 shows the dimensions of the designed QHC. The multi-branch line QHC is notorious for very narrow branch lines resulting in difficulties in accurate fabrication. In our design of QHC, the smallest line width (*W*_1_) is 0.36 mm, which can be realized using standard PCB fabrication techniques.

Figure 14a shows the reflection and transmission coefficients of the designed QHC. The reflection coefficient (S1,1) at the input Port 1 is less than −17 dB at 1.07–1.75 GHz and is −28.9 dB and −26.2 dB at 1.16 GHz and 1.61 GHz, respectively. The transmission from the input port to the isolated port (S4,1) closely tracks the reflection coefficient. The transmission from the input port to the in-phase port (S2,1) and to the quadrature-phase port (S3,1) is −3.80 dB and −3.44 dB at 1.16 GHz (0.36 dB imbalance) and −3.54 dB and −3.97 dB at 1.61 GHz (0.43 dB imbalance). Figure 14b shows the phase difference between the in-phase output and the quadrature phase output (Arg (S2,1)–Arg (3,1)). The transmission phase difference is 89.3° and 90.8° at 1.16 GHz and 1.61 GHz, respectively (±0.8° imbalance).

## 7. Design of a Crossed-Dipole Multi-GNSS Antenna

The final design of the proposed antenna for multi-GNSS applications involves combining the designs of the crossed dipole and the QHC feed network and connecting coaxial cables to the QHC ports. The combined structure is finely tuned to compensate for the interaction between the crossed dipole and the feed network, and the dimensions of the proposed antenna are slightly different from the ones in Table 3 and Table 4. Figure 15 shows a 3D view of the final design of the proposed antenna. Conducting materials are rendered in gray, while dielectric materials are in yellow.

Figure 16 shows the reflection coefficient (S1,1 for RHCP and S2,2 for LHCP) and transmission coefficient (S2,1 = S1,2) of the designed antenna. The reflection coefficient is less than –15 dB at 1.01–1.72 GHz. The transmission coefficient is less than −15 dB at 1.02–1.61 GHz, reflecting the performance of the QHC feed network. As can be seen in Figure 10, the isolation between the crossed dipoles without the feed network is greater than 24 dB at this frequency range. The limited isolation between the RHCP and LHCP ports needs to be improved if GNSS and satellite phones are to operate simultaneously using the designed antenna.

The gain and axial ratio performance of the designed antenna are presented below in rectangular plots. Figure 17 shows the gain patterns in φ = 0°, 45°, 90°, and 135° planes at 1.1, 1.3, and 1.6 GHz. The gain at zenith (*θ* = 0°) is 5.8, 5.0, and 3.9 dBic at 1.1, 1.3, and 1.6 GHz, respectively. The front-to-back ratio is in excess of 30 dB at all frequencies. The designed antenna boasts wide-beamwidth symmetric gain patterns. Analysis of the gain patterns in the horizontal plane (*θ* = 90° and −180° ≤ *φ* ≤ 180°) reveals the following values for the maximum and minimum gains:

At 1.16 GHz, *G*_max_ = −2.42 dBic at *φ* = 144° and *G*_min_ = −3.31 dBic at *φ* = 58°;At 1.30 GHz, *G*_max_ = −1.19 dBic at *φ* = 123° and *G*_min_ = −3.17 dBic at *φ* = −59°;At 1.61 GHz, *G*_max_ = −0.23 dBic at *φ* = −128° and *G*_min_ = −1.87 dBic at *φ* = 6°.

Figure 18 shows the axial ratio patterns of the designed antenna at 1.16, 1.30, and 1.61 GHz in *φ* = 0°, 45°, 90°, and 135° planes. The axial ratio is smallest at zenith and 1.0, 0.6, and 0.6 dB at 1.16, 1.30, and 1.61 GHz. In the hemispherical space, the axial ratio is less than 6 dB for all *φ* angles. The axial ratio is greatest at *θ* = ± 90°. Detailed analysis shows the following data for the axial ratio in the horizontal plane.

At 1.1 GHz, AR_max_ = 5.94 dB at *φ* = 119° and AR_min_ = 2.52 dB at *φ* = 179°;At 1.3 GHz, AR_max_ = 5.67 dB at *φ* = 45° and AR_min_ = 0.35 dB at *φ* = 76°;At 1.6 GHz, AR_max_ = 4.88 dB at *φ* = 121° and AR_min_ = 0.59 dB at *φ* = 159°.

The performance of the proposed antenna is compared with previous works in Table 5, focusing on wide beamwidth performance such as the 6 dB axial ratio (AR) beamwidth, the gain, and the AR in the horizontal plane. For the sake of fairness, we compare our design with existing works that employ crossed dipoles in Table 5. The beamwidth for AR < 6 dB is in the elevation direction for all 360-degree azimuth angles in the case of our work, while existing works seldom show the axial ratio for all azimuth angles. The frequency range is not considered critical since we focus on wide-beamwidth performance. The gain and axial ratio (AR) at *θ* = 90° are critical parameters for wide-beamwidth antenna comparison. The antenna dimension is also considered non-critical. Dual-circularly polarized antenna design is not studied in the works compared in Table 5, while our design offers an advantage of dual-polarized operation for *L*-band satellite phones as well as multi-GNSS signal reception.

The most challenging design task is to obtain a wide AR beamwidth over a large frequency range. Most of the existing work focuses on wide-beam AR design over a narrow frequency range. The work in [22] is comparable to our design in bandwidth and beamwidth performance with a smaller size. But their structure is significantly more complicated than the proposed design. The result in [23] boasts impressive beamwidth performance and gain/AR in the horizontal plane with narrower bandwidth and larger dimensions than our design.

## 8. Conclusions

We have presented a detailed description of a low-cost wide beamwidth antenna for multi-GNSS applications useful for maneuvering platforms. Wide beamwidth performance has been achieved by employing printed crossed dipoles combined with parasitic monopoles fed by a broadband quadrature hybrid coupler (QHC) and by using a ground plane of optimum diameter. The proposed antenna exhibits the following characteristics at 1.16–1.61 GHz: reflection coefficient of less than −17 dB, zenith gain of 3.9–5.8 dBic, horizontal gain of −3.3 to −0.2 dBic, zenith axial ratio of 0.6–1.0 dB, and horizontal axial ratio of 0.4–5.9 dB. In a nutshell, the gain of the proposed antenna is greater than −3.3 dBic, and the axial ratio is less than 6 dB over the entire angles of the upper hemisphere. The antenna has been realized in a dimension of 0.48 × 0.48 × 0.25 *λ* at the mid-frequency. The proposed antenna offers the capability of LHCP *L*-band satellite phone operation at 1.525–1.661 GHz. The competitive features of the proposed antenna, such as simple low-cost structure, broad bandwidth, and wide beamwidth, could develop into commercialization of a multi-GNSS antenna for maneuvering platforms.

## Figures and Tables

**Figure 1 sensors-25-04665-f001:**
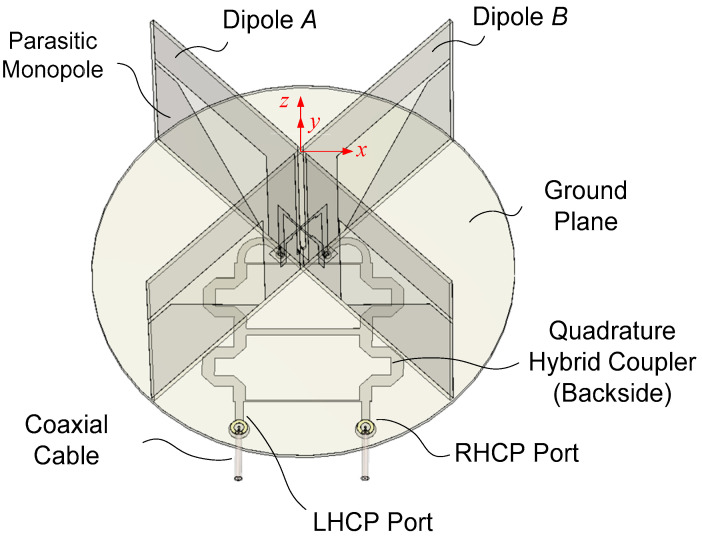
A 3D representation of the proposed antenna.

**Figure 2 sensors-25-04665-f002:**
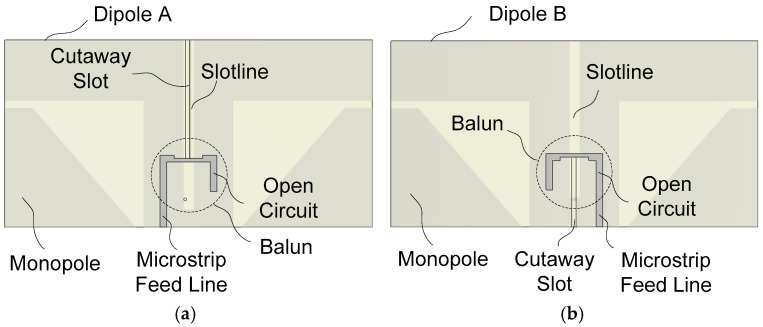
Dipole feed line with an integrated balun for (**a**) Dipole A and (**b**) Dipole B.

**Figure 3 sensors-25-04665-f003:**
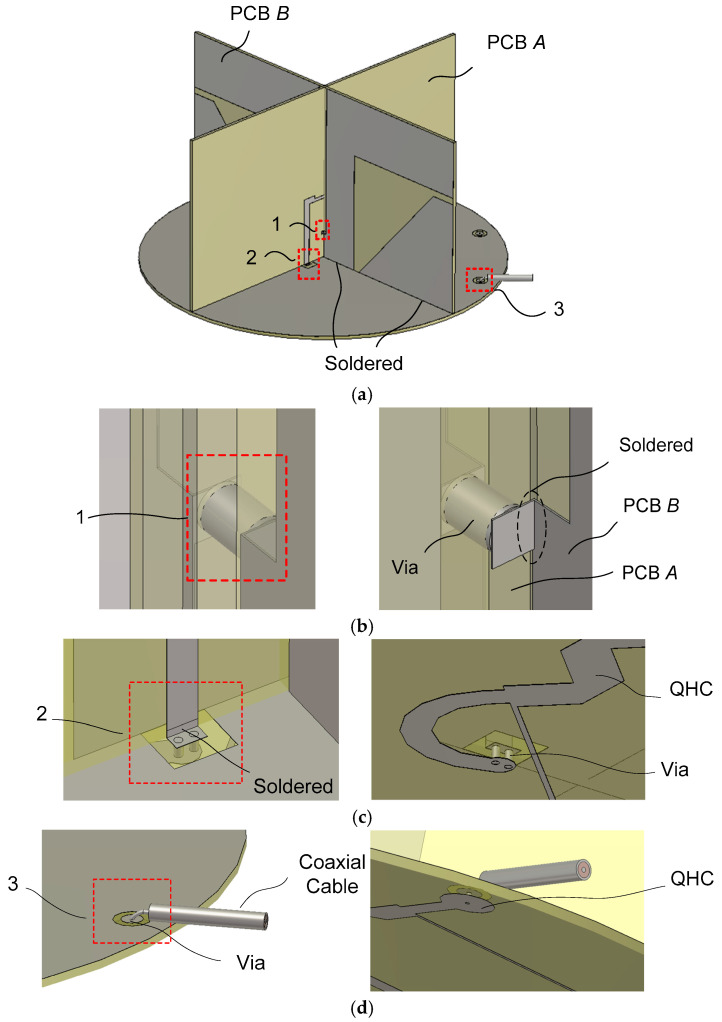
Detailed view of connecting antenna elements: (**a**) 3D view; (**b**) short-circuiting Dipole *B*’s slotline; (**c**) connecting the QHC to the dipole feed line; (**d**) connecting the QHC to a coaxial cable.

**Figure 4 sensors-25-04665-f004:**
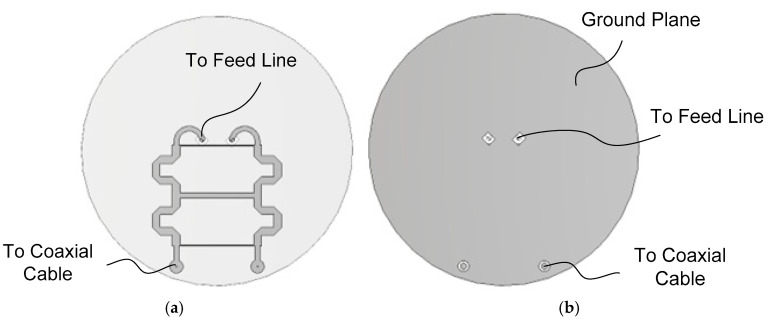
Quadrature hybrid coupler (QHC): (**a**) back-side view; (**b**) front-side view.

**Figure 5 sensors-25-04665-f005:**
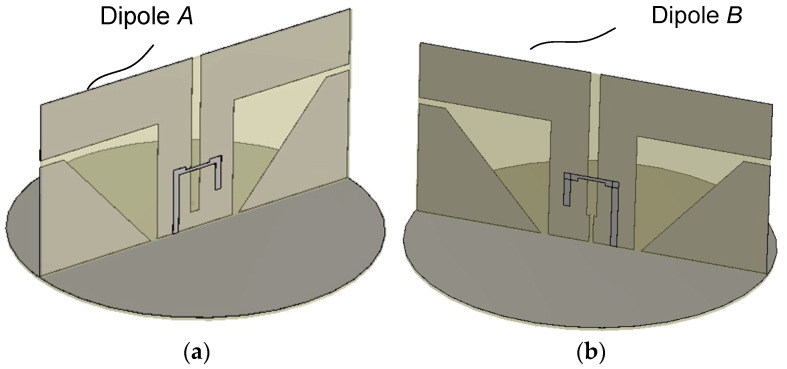
Structure for the design of a single dipole antenna. (**a**) Dipole *A* and (**b**) Dipole *B*.

**Figure 6 sensors-25-04665-f006:**
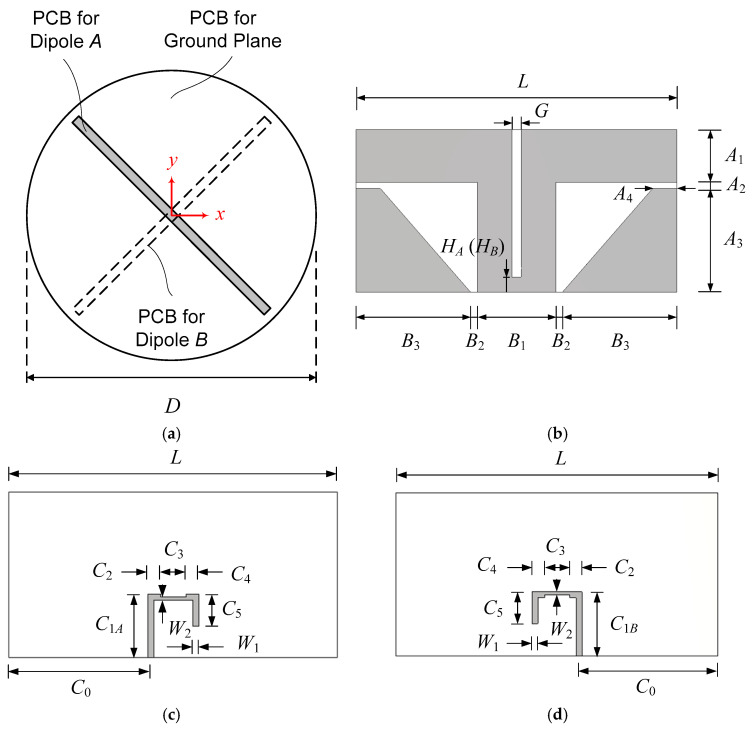
Dimensional parameters of the dipole and the feed line: (**a**) ground plane and dipole placement; (**b**) dipole and monopole; (**c**) feed line for Dipole *A*; (**d**) feed line for Dipole *B*.

**Figure 7 sensors-25-04665-f007:**
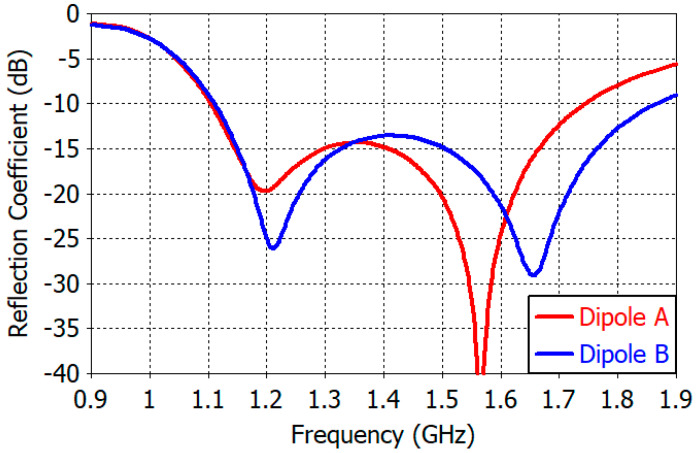
Reflection coefficient of the single dipole antenna.

**Figure 8 sensors-25-04665-f008:**
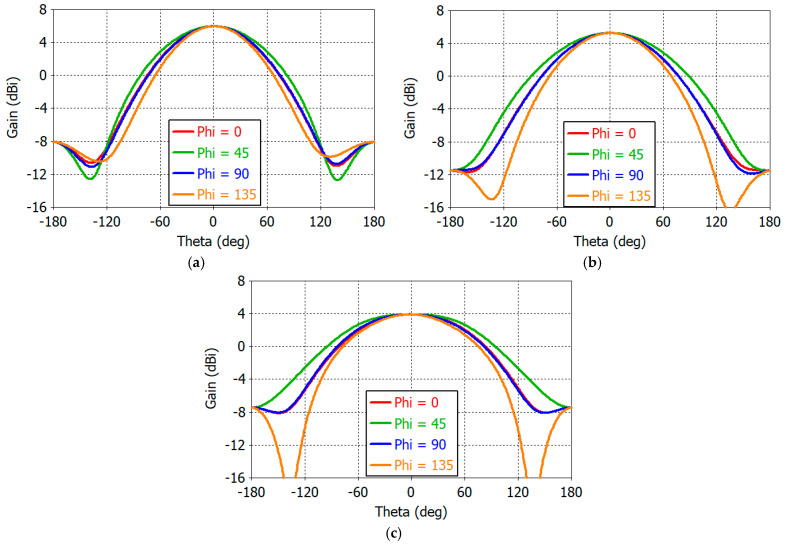
Co-polarized gain patterns of a single dipole antenna at (**a**) 1.16 GHz, (**b**) 1.30 GHz, and (**c**) 1.61 GHz.

**Figure 9 sensors-25-04665-f009:**
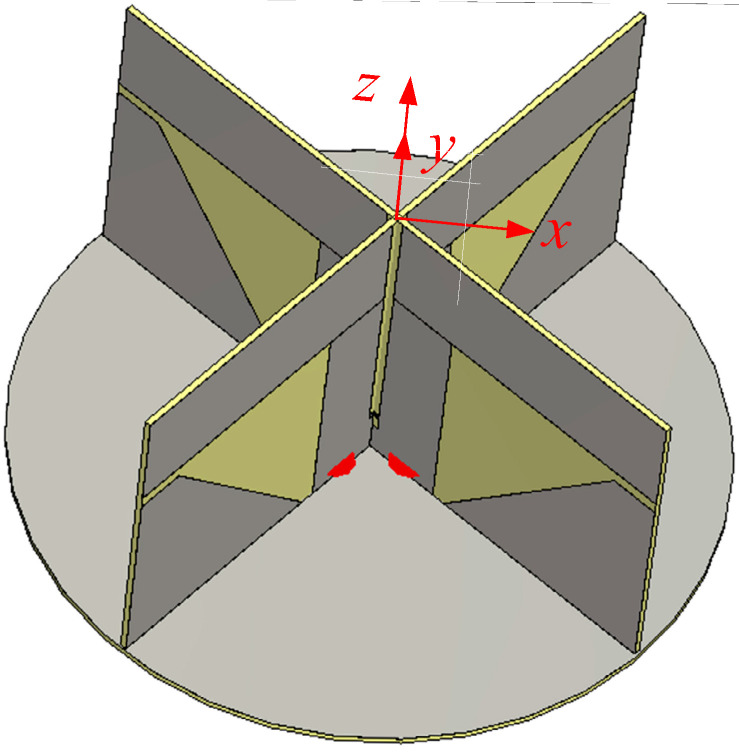
Crossed-dipole antenna with delta-gap sources.

**Figure 10 sensors-25-04665-f010:**
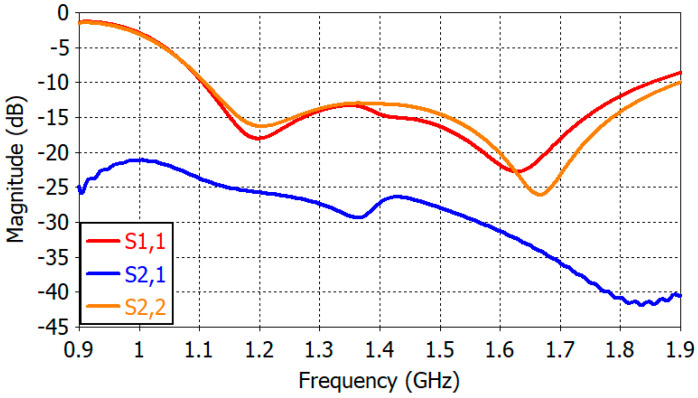
Reflection and transmission coefficients of the crossed-dipole antenna.

**Figure 11 sensors-25-04665-f011:**
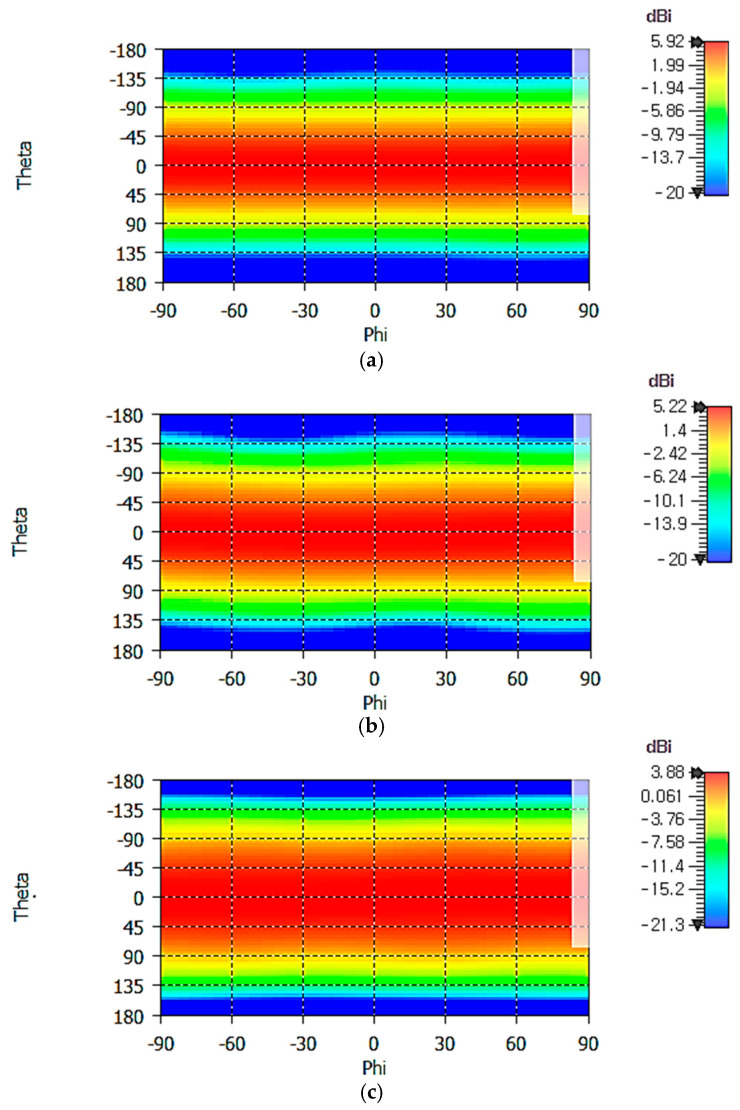
RHCP gain patterns at (**a**) 1.16 GHz, (**b**) 1.30 GHz, and (**c**) 1.61 GHz.

**Figure 12 sensors-25-04665-f012:**
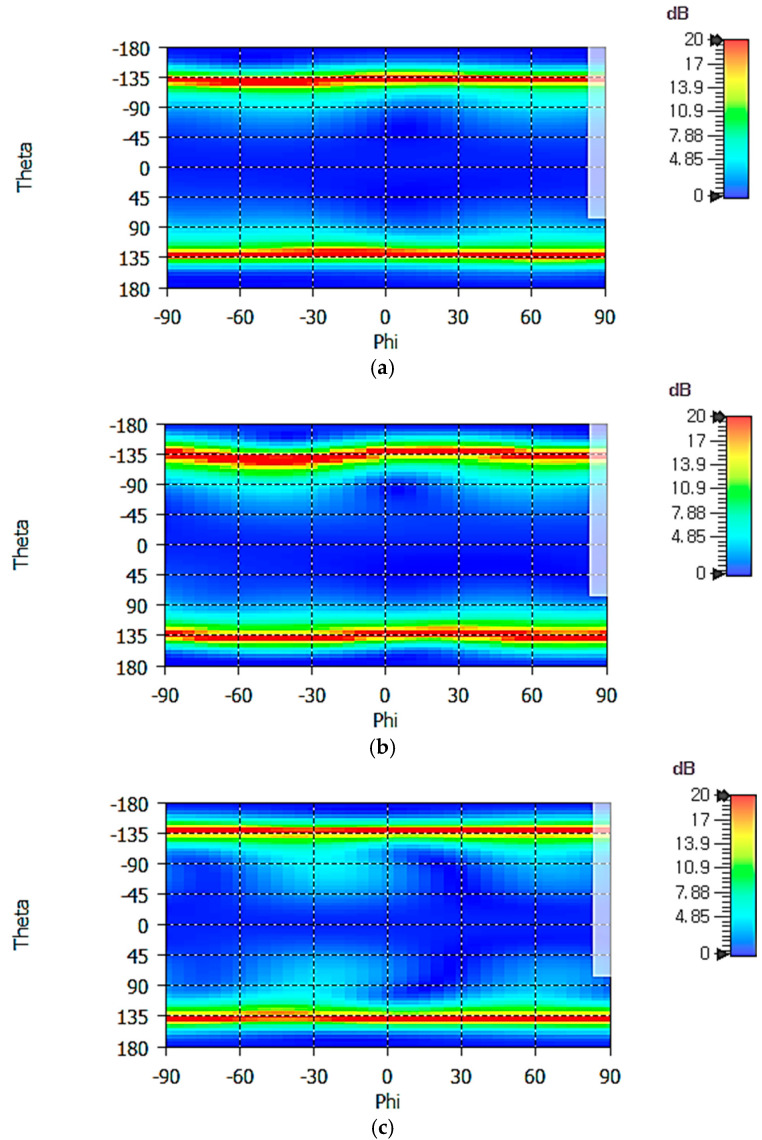
Axial ratio patterns of the crossed-dipole antenna at (**a**) 1.16 GHz, (**b**) 1.30 GHz, and (**c**) 1.61 GHz.

**Figure 13 sensors-25-04665-f013:**
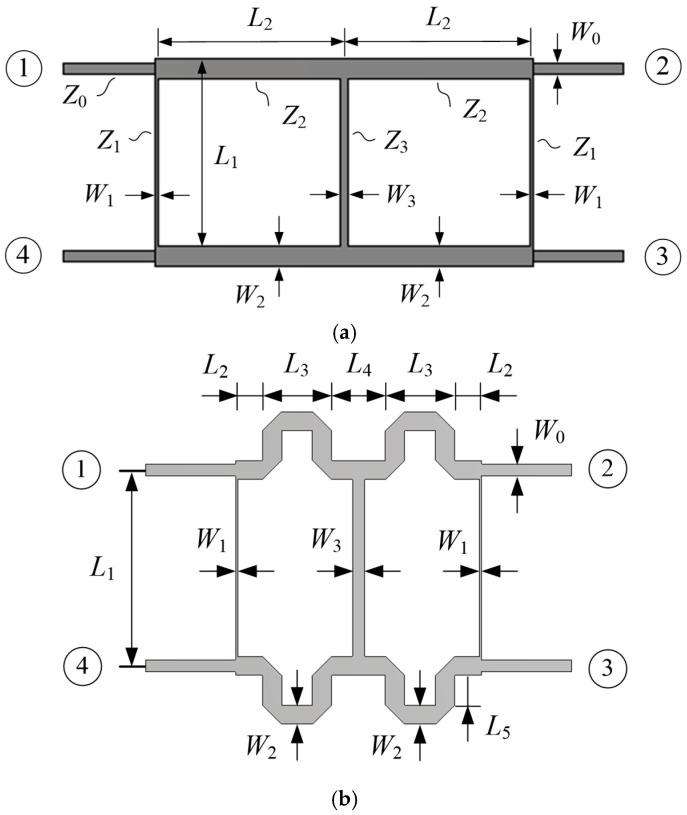
Structure of a three-branch line coupler (**a**) before size reduction and (**b**) after size reduction.

**Figure 14 sensors-25-04665-f014:**
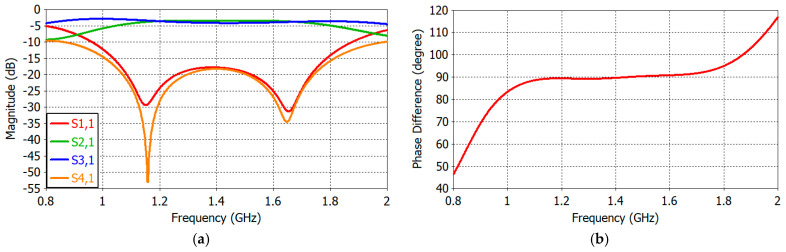
(**a**) Reflection and transmission coefficients and (**b**) the phase difference between two output ports of the designed QHC.

**Figure 15 sensors-25-04665-f015:**
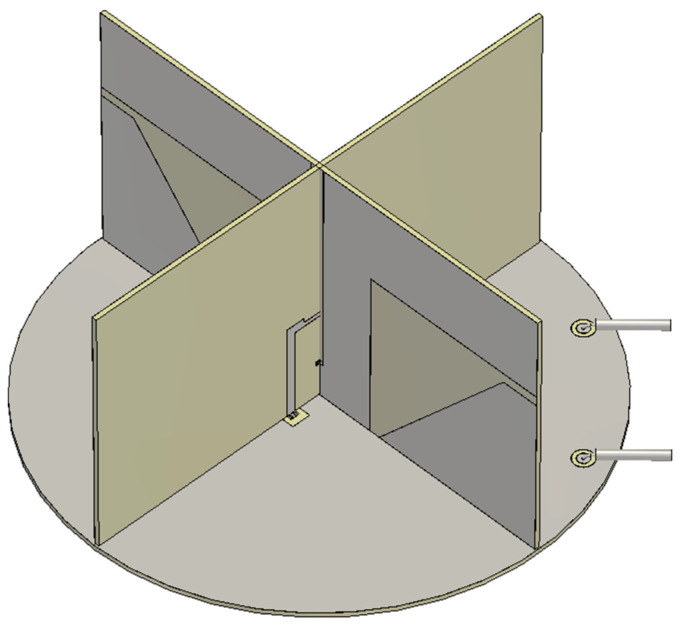
A 3D view of the designed antenna.

**Figure 16 sensors-25-04665-f016:**
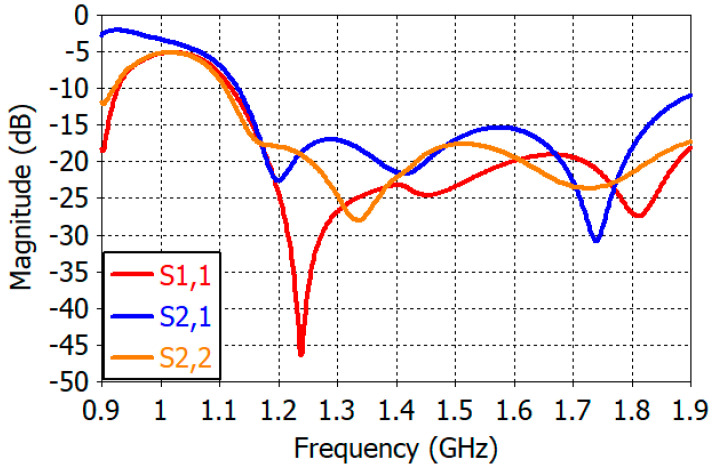
Reflection coefficient and isolation characteristics of the designed antenna.

**Figure 17 sensors-25-04665-f017:**
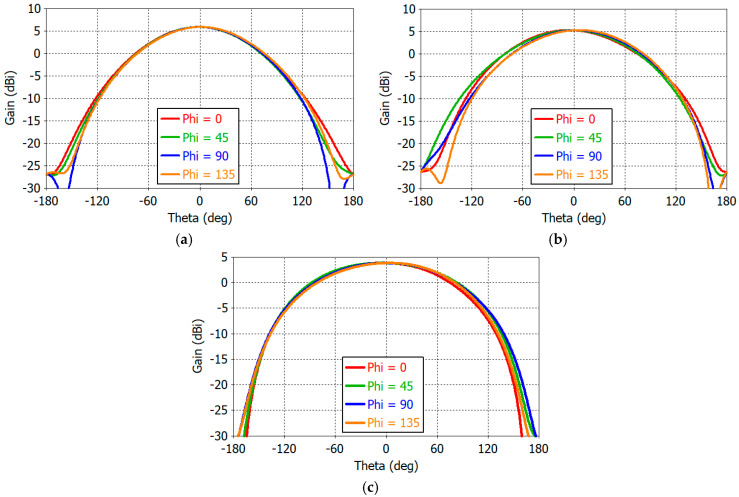
Gain patterns of the designed antenna at (**a**) 1.16 GHz, (**b**) 1.30 GHz, and (**c**) 1.61 GHz.

**Figure 18 sensors-25-04665-f018:**
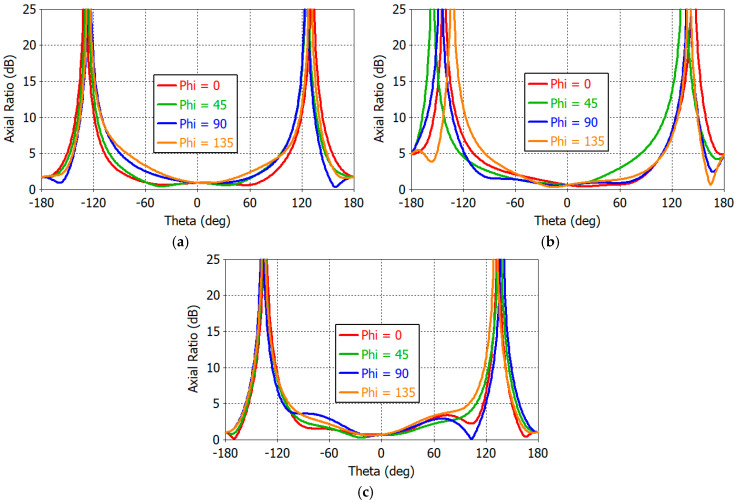
Axial ratio patterns of the designed antenna at (**a**) 1.1 GHz, (**b**) 1.3 GHz, and (**c**) 1.6 GHz.

**Table 1 sensors-25-04665-t001:** Design specifications of the proposed antenna.

Parameter	Specification
Frequency range (GHz)	1.16–1.61
Input reflection coefficient (dB)	<−10
Polarization	RHCP
Gain at zenith (dBic)	>3
Horizontal gain (dBic)	>−4
Axial ratio (dB)	<6 (in the upper hemisphere)
Size (*λ* at center freq.)	<0.5 × 0.5 × 0.3

**Table 2 sensors-25-04665-t002:** Axial ratio versus the magnitude ratio and phase difference in two vector components.

Axial Ratio (dB)				
Arg(*E_θ_*) − Arg(*E_φ_*)	|*E_θ_*|/|*E_φ_*| (dB)			
	0	1	2	3
90	0.00	1.00	2.00	3.00
85	0.76	1.26	2.14	3.10
80	1.52	1.83	2.52	3.38
75	2.30	2.51	3.06	3.81
70	3.09	3.26	3.71	4.35
65	3.92	4.05	4.43	4.99
60	4.77	4.88	5.21	5.71
55	5.67	5.77	6.06	6.51
50	6.63	6.72	6.98	7.39

**Table 3 sensors-25-04665-t003:** Dimensions of the designed antenna (mm).

Parameters	Value	Parameters	Value	Parameters	Value
*A* _1_	17.50	*C* _0_	45.80	*D*	107.00
*A* _2_	2.00	*C* _1*A*_	20.72	*H_A_*	5.00
*A* _3_	34.40	*C* _1*B*_	21.22	*H_B_*	7.50
*A* _4_	8.00	*C* _2_	4.11	*G*	3.00
*B* _1_	26.00	*C* _3_	8.22	*L*	106.10
*B* _2_	2.20	*C* _4_	4.11	*W* _1_	1.94
*B* _3_	37.85	*C* _5_	10.44	*W* _2_	0.97

**Table 4 sensors-25-04665-t004:** Dimensions of the designed QHC (mm).

Parameters	Value	Parameters	Value	Parameters	Value
*L* _1_	31.72	*L* _4_	8.70	*W* _1_	0.36
*L* _2_	4.35	*L* _5_	4.80	*W* _2_	3.10
*L* _3_	11.20	*W* _0_	1.94	*W* _3_	1.94

**Table 5 sensors-25-04665-t005:** Comparison with previous works.

Ref.	AR (<6 dB) Beamwidth (deg)	Freq. Range (GHz)	Gain (dBic) Min. at *θ* = 90°	AR (dB) Max. at *θ* = 90°	Dimension (*λ*^3^) *λ* at Center Freq.	Dual Circular Polarization
[22]	170	1.227–1.575	−3.3	7.0	0.32 × 0.32 × 0.17	No
[23]	245	2.075	−4.2	3.8	1.05 × 1.05 × 0.55	No
[24]	185	0.29–0.40	−6.6	3.3	0.35 × 0.35 × 0.19	No
[25]	190	1.545–1.625	−2.7	5.0	0.85 × 0.85 × 0.42	No
This Work	180	1.10–1.61	−3.3	5.9	0.48 × 0.48 × 0.25	Yes

## Data Availability

The data presented in this study are available in this article.

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
