# Peer review of "Simulation-Based Design of a Low-Cost Broadband Wide-Beamwidth Crossed-Dipole Antenna for Multi-Global Navigational Satellite System Positioning"

_sensors, 2025, doi:10.3390/s25154665_

Round 1
Reviewer 1 Report
Comments and Suggestions for Authors
- On the page 2 line 49 author state that “it is important to maintain a minimum gain…” Could authors explain it? Usually, it is preferably to attain maximum gain.
- On the page 2 line 51 sentence “A minim as…” seems uncompleted.
- Measurement unit “dBic” should be explained. What does “c” mean?
- Design specification of the proposed antenna in table 1 should be substantiated. Why do the authors use these parameters to synthesize the antenna? What are the most popular prototypes of such antennas? Is this being a niche for amateurs’ low-cost antenna?
- Sections 3-6 describe simulation-based design. It is well described. However, the material is perceived as a computational graphic work in software with interpretation of the results. It is obvious, that the work has some practical significance. However, the theoretical significance and scientific novelty remain unclear. What are the theoretical guidelines for the search? If it is a tutorial for “Simulation-Based Design…”, it is OK. If the manuscript is termed “Article”, in the reviewer's opinion, the authors should add some theoretical justification. Without this, the work is perceived as an analysis of the results of modeling when working in CST Studio Suite special software. In the reviewer's opinion this is more of an engineering search than a scientific article.
- Table 4. “Comparison with previous works” is good, however, some theoretical guidelines for the search seem necessary to add.
Author Response
Authors' to Reviewer 1's Commnents
We appreciate the reviewer's comments, which helped a lot in improving our manuscript. The following is our response to the reviewer's comments.
- On the page 2 line 49 author state that “it is important to maintain a minimum gain…” Could authors explain it? Usually, it is preferably to attain maximum gain.
(Response)
We have revised the text as follows.
"In designing, GNSS antennas with a wide beamwidth on board highly maneuvering platforms, it is important to maintain required gain and axial-ratio (AR) from the zenith down to the horizon for reliable reliable reception of GNSS signals."
- On the page 2 line 51 sentence “A minim as…” seems uncompleted.
(Response)
We have corrected the error.
- Measurement unit “dBic” should be explained. What does “c” mean?
(Response)
It means dB gain relative to isotropically radiating circularly polarized antenna. We have added:
"A circularly polarized antenna radiating isotropically in all angles has a gain of 0 dBic, where the subscript 'ic' means 'relative to isotropic circularly polarized radiator'."
- Design specification of the proposed antenna in table 1 should be substantiated. Why do the authors use these parameters to synthesize the antenna? What are the most popular prototypes of such antennas? Is this being a niche for amateurs’ low-cost antenna?
1) Design specification of the proposed antenna in table 1 should be substantiated. Why do the authors use these parameters to synthesize the antenna?
(Response)
Please note that we have tried to explain the logic behind numbers in Table 1 (Lines 110-121 in the revised manuscript).
"For impedance-matched performance, the reflection coefficient of less than −10 dB is specified. For wide beamwidth operation, the minimum antenna gain at zenith (θ = 0°) needs to be relaxed to 3 dBic. The minimum antenna gain at horizon (θ = 90°) is set to −4 dBic reflecting the physical realizability in a given dimension. An omnidirectional gain of 3 dBic over the hemispherical space and no radiation (minus infinite gain) below the horizon can be obtained only from an ideal imaginary antenna, which occupies an infinitely large space. The axial ratio of less than 6 dB is specified over the upper hemispherical angles (−90° ≤ θ ≤ 90° and −90° ≤ φ ≤ +90°) not just on the principal planes. This value is also selected reflecting the realizability. The normal axial ratio of 3 dB is difficult to achieve at all angles in the upper hemisphere so that it is relaxed to 6 dB. Assuming a 1-dB axial ratio for the GNSS transmitting antenna onboard a satellite, the former corresponds to 0.23 dB in the maximum polarization mismatch loss while the latter to 0.64 dB [27]."
Upon reviewer's comment, we have added:
"The first six parameters from frequency range to axial ratio are essential performance criteria required of a GNSS antenna. Antenna size is typical of a single antenna that can meet design specifications in Table 1. GNSS RF signals are transmitted in the RHCP (right hand circular polarization)."
2) What are the most popular prototypes of such antennas?
(Response) We have added:
"Antenna types capable of meeting design specifications in Table 1 include crossed-dipoles [22−25], four-point fed patches [26], four horizontal arc monopoles [27], and quadrifilar helices [28]."
3) Is this being a niche for amateurs’ low-cost antenna?
(Response) We have added:
"Antenna types capable of meeting design specifications in Table 1 include crossed-dipoles [21−26], four-point fed patches [28], four horizontal arc monopoles [29], and quadrifilar helices [30]. In this work, we choose wideband crossed-dipoles with parasitic monopoles fed by a broadband quadrature hybrid coupler realized on the dipole ground plane of optimum size. This structure requires only three printed circuit boards (PCB's) that can be easily assembled by inserting PCB's in slots and using soldering PCB's at suitable locations. The proposed antenna with a wide beamwidth is amenable to low-cost production and, for example, can be attractive for small fishing boats that can constantly undergo rolling and pitching."
- Sections 3-6 describe simulation-based design. It is well described. However, the material is perceived as a computational graphic work in software with interpretation of the results. It is obvious, that the work has some practical significance. However, the theoretical significance and scientific novelty remain unclear. What are the theoretical guidelines for the search? If it is a tutorial for “Simulation-Based Design…”, it is OK. If the manuscript is termed “Article”, in the reviewer's opinion, the authors should add some theoretical justification. Without this, the work is perceived as an analysis of the results of modeling when working in CST Studio Suite special software. In the reviewer's opinion this is more of an engineering search than a scientific article.
(Response) Upon reviewer's comment, we have added some theory on the circular polarization (CP), the axial ratio (AR) and the design of CP antennas with good AR using the proposed structure. We have added some equations and a table. Please refer to the revised manuscript.
- Table 4. “Comparison with previous works” is good, however, some theoretical guidelines for the search seem necessary to add.
(Response) We have added:
"For the sake of fairness, we compare our design with existing works that employ crossed dipoles in Table 5. The beamwidth for AR < 6 dB is in the elevation direction for all 360-degree azimuth angles in the case of our work, while existing works seldom show the axial ratio for all azimuth angles. The frequency range is not considered critical since we focus on wide-beamwidth performance. The gain and axial ratio (AR) at θ = 90° are critical parameters for wide-beamwidth antenna comparison. The antenna dimension is also considered non-critical. Dual-circularly polarized antenna design is not studied in the works compared in Table 5, while our design offers an advantage of dual-polarized operation for L-band satellite phone as well as multi-GNSS signal reception."

Reviewer 2 Report
Comments and Suggestions for Authors
1. 50 remove a text.
2. 99 "2. Design Specifications and Antenna Structure" can be 2 separate chapters.
3. 187- 255 The slots for assembling of two PCB as the cross structure should be described .
4. 233 Add marks for the which point a, b or c in the Figure 3 a), magnified pictures corresponds.
5. 331 Since the dipoles are slightly different, the S11 of the both dipoles can be presented in the Figure 7.
6. 268 In the Figure 6 the same dipole is presented in the both sides.
7. 336 What gain (Total, X, E....) is presented in Figure 8?
8. Clarify and display on antenna's picture, what are the angles theta and phi in the Figure 8 and further?
9. Why typical dipole minimal radiation is not visible at opposite directions of azimuth?
10. 358 "Figure 12" should be replaced to "Figure 11"
11. 379 "This can be seen by comparing Figure 12a with Figure 12c. The AR is largest at horizon (θ = 90°)" According to the Figure 12, max. axial ratio is 135 degrees. Explain please.
12. To avoid confusion with definitions of dimensions, 1D and 2D should be replaced
13. 354 S12 is invisible. You can manipulate the width of the line.
Author Response
Author's response to Rreviewer 2's Comments
We appreciate reviewer's constructive comments. We have earnestly tried to correct the manuscript according to reviewer's comments.
- 50 remove a text.
(Response) We have corrected it.
- 99 "2. Design Specifications and Antenna Structure" can be 2 separate chapters.
(Responsee) We corrected it to: "2. Design Specifications; 3. Antenna Structure, and so on."
- 187- 255 The slots for assembling of two PCB as the cross structure should be described .
(Response) We have revised Figure 2. We have added:
"To assemble two dipole PCB's in a cross configuration, a cutaway slot is formed on the Dipole A PCB and on the Dipole B PCB. The width of the cutaway slot is slightly larger than the thickness of the PCB. The cutaway slot ranges from just above the microstrip line to the top edge in the Dipole PCB A while it ranges from the bottom edge to just below the microstrip line."
- 233 Add marks for the which point a, b or c in the Figure 3 a), magnified pictures corresponds.
(Response) To make it clear, we add notations a, b, and c in Figures 3 b), 3 c), and 3 d).
- 331 Since the dipoles are slightly different, the S11 of the both dipoles can be presented in the Figure 7.
(Response) We have added the reflection coefficient of the Dipole B in Figure 7.
- 268 In the Figure 6 the same dipole is presented in the both sides.
(Response) We have corrected the error.
- 336 What gain (Total, X, E....) is presented in Figure 8?
(Response) It is the co-polarized gain. We have corrected the text and the caption of Figure 8.
- Clarify and display on antenna's picture, what are the angles theta and phi in the Figure 8 and further?
(Response) We have added coordinate system axes in Figure 9. We have added the following text:
" Rectangular coordinate axes are shown in Figure 9 for far-field angle identification in spherical coordinates. Dipoles A and B are placed in φ = 45° and φ = 135° planes, respectively."
- Why typical dipole minimal radiation is not visible at opposite directions of azimuth?
(Response) Antenna gain patterns in Figure 11 is the one generated by two crossed dipoles fed in phase quadrature resulting in almost omnidirectional gain in the azimuth direction.
- 358 "Figure 12" should be replaced to "Figure 11"
(Response) We have corrected the error.
- 379 "This can be seen by comparing Figure 12a with Figure 12c. The AR is largest at horizon (θ = 90°)" According to the Figure 12, max. axial ratio is 135 degrees. Explain please.
(Response) We meant the largest AR in the upper hemisphere. We have added: "The AR is largest at horizon (θ = 90°) in the upper hemisphere (−90° ≤ θ ≤ +90°)."
- To avoid confusion with definitions of dimensions, 1D and 2D should be replaced
(Response) We have removed 1D in the Figure 8 caption, and 2D in the caption of Figures 11 and 12.
- 354 S12 is invisible. You can manipulate the width of the line.
(Response) We have removed S1,2 in the caption and added a text: "The transmission coefficient S1,2 from Dipole B to Dipole A is not drawn in Figure 10 since it is the same as S2,1."

Round 2
Reviewer 1 Report
Comments and Suggestions for Authors
Please add in table 2 or below some explanations for terms 0, 1, 2, 3 in row 3. Assessing the work as a whole, the reviewer is satisfied with the authors' responses and revisions and believes that it can be accepted for publication.
Author Response
Response to Reviewer 1, 2nd Round
Comment 1: Please add in table 2 or below some explanations for terms 0, 1, 2, 3 in row 3. Assessing the work as a whole, the reviewer is satisfied with the authors' responses and revisions and believes that it can be accepted for publication.
Response 1: We appreciate reviewer's comment. We have added the following in Lines 172-174, above Table 2.
The axial ratio is calculated and tabulated using Equation 6 for the phase difference (Arg(Eθ)−Arg(Eφ)) of 50 to 90 degrees and the magnitude ratio (|Eθ|/|Eφ|) of 0, 1, 2, and 3 dB.